# Genetic Diversification and Selection Strategies for Improving Sorghum Grain Yield Under Phosphorous-Deficient Conditions in West Africa

**Chiaka Diallo [1], H. Frederick W. Rattunde [2,*], Vernon Gracen [3], Aboubacar Touré [4], Baloua Nebié [4], Willmar Leiser [5]** 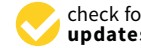 **, Daniel K. Dzidzienyo [6], Ibrahima Sissoko [4], Eric Y. Danquah [6], Abdoulaye G. Diallo [7], Bakary Sidibé [4], Mamourou Sidibé [4] and Eva Weltzien [2]**

[1] Department of Agronomic Sciences, Institut Polytechnique Rural de Formation et Recherche Appliquée de Katibougou, (IPR/IFRA), BP 06 Koulikoro, Mali; dchiaka@77yahoo.fr

[2] Agronomy Department, University of Wisconsin-Madison, Madison, WI 53706, USA; eva.weltzien@gmail.com

[3] School of Integrative Plant Sciences, Section of Plant Breeding and Genetics Cornell University, Ithaca, NY 14853, USA; vg45@cornell.edu

[4] Sorghum Program, International Crops Research Institute for Semi-Arid Tropics, (ICRISAT), 65 Bamako, Mali; ab.toure@cgiar.org (A.T.); b.nebie@cgiar.org (B.N.); i.sissoko@icrisatml.org (I.S.); b.sidibes@icrisatml.org (B.S.); M.Sidibe@cgiar.org (M.S.)

[5] State Plant Breeding Institute, Univ. of Hohenheim, 70599 Stuttgart, Germany; willmar_leiser@uni-hohenheim.de

[6] West Africa Center for Crop Improvement (WACCI), University of Ghana (UG), P.O. Box LG 25 Accra, Ghana; ddzidzienyo@wacci.ug.edu.gh (D.K.D.); edanquah@wacci.ug.edu.gh (E.Y.D.)

[7] Institut d'Economie Rurale, BP 262 Bamako, Mali; ag_diallo@hotmail.com

\* Correspondence: f.rattunde@gmail.com

**Abstract:** Sorghum, a major crop for income generation and food security in West and Central Africa, is predominantly grown in low-input farming systems with serious soil phosphorus (P) deficiencies. This study (a) estimates genetic parameters needed to design selection protocols that optimize genetic gains for yield under low-phosphorus conditions and (b) examines the utility of introgressed backcross nested association mapping (BCNAM) populations for diversifying Malian breeding materials. A total of 1083 BC1F5 progenies derived from an elite hybrid restorer "Lata-3" and 13 diverse donor accessions were evaluated for yield and agronomic traits under contrasting soil P conditions in Mali in 2013. A subset of 298 progenies were further tested under low-P (LP) and high-P (HP) conditions in 2014 and 2015. Significant genetic variation for grain yield was observed under LP and HP conditions. Selection for grain yield under LP conditions was feasible and more efficient than the indirect selection under HP in all three years of testing. Several of the BCNAM populations exhibited yields under LP conditions that were superior to the elite restorer line used as a recurrent parent. The BCNAM approach appears promising for diversifying the male parent pool with introgression of diverse materials using both adapted Malian breed and unadapted landrace material from distant geographic origins as donors.

**Keywords:** base broadening; germplasm; sorghum; phosphorus adaptation; selection strategy

## 1. Introduction

Sorghum is one of the most important crops for smallholder farmers in West Africa who annually cultivate 14.1 million ha, approximately half of African and one-third of world production area of sorghum [1]. This cereal crop is produced in low-input farming systems [2–4] in which soil phosphorous

deficiency is widespread and a serious constraint to yield [5,6]. Sorghum is critical for food security and plays an important role in West African farming systems as it can be produced under poor soil conditions [7]. Nevertheless, soil phosphorous (P) deficiency limits the growth and development of sorghum, reducing grain yield and delaying maturity, which increases exposure to end-of-season drought stress [6,8]. Specific efforts to breed sorghum with superior grain yield under P-limited conditions are necessary to optimize genetic gains for grain yield in these environments [6,9]. Genetic variation occurs for yields under low P-fertility [6] and requires appropriate statistical designs to reliably determine grain yield in low-P fields [6]. These findings were based on a group of pure-line varieties that do not represent the situation when selecting in elite segregating materials for developing superior hybrid parents.

Sorghum breeding in Mali and Burkina Faso emphasizes the use of Guinea-race germplasm, the race predominantly cultivated in West Africa. Random-mating populations based on West African Guinea landrace varieties have been created as a source of diversity [7,10] and subjected to several cycles of selection for grain yield and key traits for adaptation and farmer acceptability. This work produced novel, dual-purpose (grain plus fodder) varieties with desirable Guinea-race grain, but the genetic gains for grain yield were low. One factor hindering gains progress for grain yield could have been the low genetic diversity within West African Guinea-race varieties. The low genomic diversity found within West African Guinea-race varieties, relative to global diversity within the Guinea race and over races [11], supports this hypothesis. An additional factor could have been low population sizes, even when starting with very large populations, because of the strong culling for numerous traits (e.g., grain hardness, free threshing glumes, panicle form sufficiently lax to minimize head-bug and grain mold losses) required for adaptation and farmer acceptance of new varieties.

The nested association mapping (NAM) method was developed to dissect complex traits in maize [12]. A modification of the NAM method using backcrossing, the backcross nested association mapping (BC-NAM), was employed in sorghum for using exotic, unadapted germplasm to increase genetic diversity while retaining critical gene complexes essential for adaptation and efficient hybrid breeding (minimizing disturbance of fertility restoration genes) [13]. An elite hybrid male was used as the recurrent parent for a single backcross generation, and the resulting BC1F1 populations were selected for plant height and maturity [14]. This method was reported to be useful for both applied breeding and genetic dissection of yield and other complex agronomic traits using molecular genetic tools [13].

A series of BCNAM populations were created in Mali using the pure line variety and successful hybrid male parent "Lata3" as the recurrent parent. The variety "Lata3" was derived from a West African Guinea-race random-mating population [7] and identified through farmer-participatory variety testing in Mali. Furthermore, several donor parents used to create these populations were useful for diversifying the hybrid male parent pool and increasing heterosis with the female pool based Guinea-race materials from Mali and neighboring countries [15,16]. Therefore, these populations represent promising materials for genetic diversification of West African Guinea-race sorghum for pure-line as well as hybrid variety development.

This study seeks to support sorghum breeding in West Africa with (1) genetic parameters needed for designing efficient breeding programs targeting yield improvement under low-P production conditions and (2) information regarding the usefulness of specific Lata3-BCNAM backcross populations and the BCNAM method for increasing genetic variation for grain yield under contrasting P fertility conditions.

## 2. Materials and Methods

### 2.1. Plant Materials

Thirteen biparental backcross populations were developed using the same recurrent parent "Lata" and 13 different donor parents (Table 1). The recurrent parent was derived from a random-mating Guinea Population [7] and identified through farmer-participatory variety testing [16,17]. Lata3 was

chosen as the recurrent parent because of its importance. It is cultivated as a novel intermediate-height, pure-line variety and used as the male parent for successful hybrids even though it has weaknesses including suboptimal glume opening and susceptibility to *Striga*. The 13 donor parents were chosen to represent geographical and racial diversity from within the Guinea-race and other races (Table 1). They were also chosen to contribute traits that contribute to adaptation and farmer acceptance of new varieties including tolerance to biotic (*Striga*, sorghum midge) and abiotic (soil phosphorus deficiency, aluminum toxicity) stresses, quality (grain vitreousness, stem sweetness), and panicle desirability (laxness, glume opening).

**Table 1.** Donor parents with racial classifications (Snowden classifications of Guinea-race), origins, number of BC1F5 progenies tested, and desirable traits of each parent.

| Donor | Race | Pop Names | Origin | Progenies | Specific Advantages |
|-------|------|-----------|--------|-----------|---------------------|
| N'golofing | Guinea (Guineense) | N'golo | Mali | 80 | Grain and panicle traits |
| Douadjè | Guinea (Guineense) | Douad | Mali | 80 | Low-P adaptation and also allele for Al tolerance. |
| Gnossiconi | Guinea (Guineense) | Gnoss | Burkina Faso | 71 | Grain and panicle traits |
| Sambalma | Guinea (Conspicuum) | Samba | Nigeria | 102 | Grain and panicle traits, Al tolerance |
| IS15401 | Guinea (Conspicuum) | Soumb | Cameroon | 101 | *Striga*, aluminum, low-P, and midge tolerance/resistance |
| Fara-Fara | Guinea (Conspicuum) | Fara | Nigeria | 80 | Geographic and intra-racial diversity |
| IS23645 | Guinea (Margaritiferum) | Hafid | Gambia | 75 | Intra-racial diversity and vitreous grain |
| Grinkan | Caudatum-Guinea | Grinka | Mali | 100 | Productivity, Stover quality |
| Framida | Caudatum | Fram | Burkina Faso | 80 | *Striga* tolerance |
| Ribdahu | Caudatum | Ribda | Nigeria | 80 | Midge resistance, racial diversity |
| SC566-14 | Caudatum | SC566 | Brazil | 80 | Aluminum tolerance |
| IS 23540 | Caudatum | IS235 | Ethiopia | 80 | Sweet stem |
| SK 5912 | Durra-Caudatum | SK591 | Nigeria | 80 | Geographic and racial diversity |

The recurrent parent was crossed to the 13 donor parents and then backcrossed to each of the resulting F1s (Figure 1) following the method described by [13]. Plants of the BC1F1 and subsequent generations were selected for heading date and plant height similar to that of the recurrent parent. From 70 to 102 progenies were advanced to the BC1F4 generation for each of the 13 backcross nested association mapping populations (BC-NAM), from which 1083 BC1F5 progenies were obtained for phenotyping (Table 1). The off-season and the rainy season were used to develop the BC1F4 progenies, but selection for heading and plant height was done only in the rainy season. These 13 backcross nested association mapping (BC-NAM) populations based on the recurrent parent Lata3 contributed to a larger set of materials, involving two other recurrent parents, developed through collaboration of three institutes: International Crops Research Institute for the Semi-Arid Tropics (ICRISAT-Mali) and Institute Economics Rural (IER-Sotuba Mali) and CIRAD Montpellier France.

A subset of 298 of the more promising BC1F5 progenies was identified from the full set of 1083 progenies for subsequent evaluations of genotype performance and genotype by P-level interaction over multiple years. This subset of progenies included the top 15% of the progenies for grain yield in 2013 under either low-P (LP) or high-P (HP) conditions ($n = 258$). An additional 40 progenies were added to the subset based on use of a selection index calculated with standardized best linear unbiased

estimates (BLUEs) of 2013 grain yield, women's appreciation of grain quality, threshability, resistance to foliar anthracnose, and photoperiod sensitivity. The economic weights used in the selection index were 0.5 for grain yield and 0.1 for each of the remaining traits.

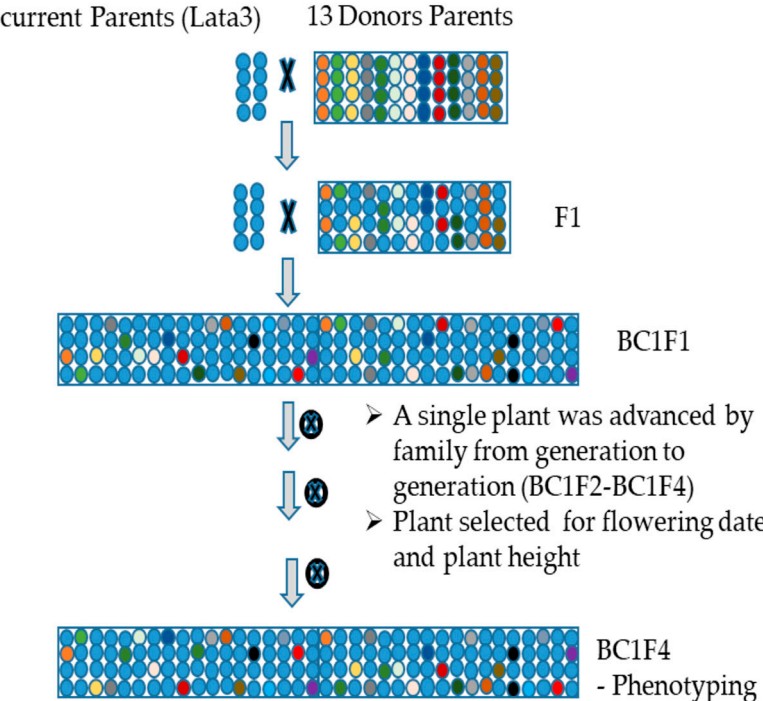

**Figure 1.** Development of the BC1F4 NAM population with Lata 3 as the recurrent parent and 13 donor parents.

## 2.2. Phenotyping

The entire set of 1083 BC1F5 progenies and check varieties were tested under both LP and HP conditions in 2013 at the ICRISAT-Samanko research station (120 31' N, 80 4' W Figure 2). Adjacent fields managed for contrasting phosphorous (P) status were chosen for these trials based on their cropping history with HP fields having sorghum–groundnut rotations with annual applications of 100 kg/ha diammonium phosphate (DAP), whereas the LP fields were fallowed multiple years prior to initiating cultivation with no inorganic P fertilization.

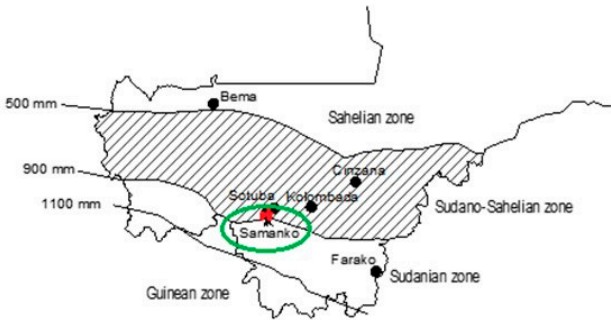

**Figure 2.** Location of Samanko, the Sudano-Sahelian zone (800–1000 mm).

The LP trials received no P fertilization, whereas the HP trials received 20 m$^{-2}$ elemental P applied in the form diammonium phosphate (DAP) at the rate of 100 kg ha$^{-1}$ prior to sowing. Both LP and HP trials received equal quantities of nitrogen (N) via topdressings of 50 kg ha$^{-1}$ urea at approximately four weeks after sowing and an additional application of 37.5 kg ha$^{-1}$ urea at sowing to the LP trials to

match the N applied to the HP trials as basal fertilizer. An alpha lattice design with incomplete blocks of 11 plots and two replications was used. Under HP, the plant available P using Bray-1, was above 14 ppm (14 mg kg$^{-1}$ soil), and under LP it was below 5 ppm (5 mg kg$^{-1}$ soil).

The 298 selected BC1F5 progenies and the recurrent parent, Lata3, were tested for grain yield at Samanko in 2014 and 2015. Adjacent fields under LP and HP conditions with management identical to 2013 were used in 2014 and 2015. An alpha design with 300 entries, incomplete blocks of 5 plots, and 3 replications was used for both LP and HP trials in each year.

The field plots consisted of a single row (2013) or two rows (2014 and 2015) 3 m in length with a 75 cm inter-row distance and 30 cm between hills. All plots were thinned to two plants per hill. Both the LP and HP trials were sown on the same day (28 June 2013; 30 June 2015) or within one day of each other (LP, 24 June 2014; HP, 25 June 2014). The cumulative rainfall was 1180, 1209, and 1367 mm at Samanko in 2013, 2014, and 2015, respectively.

Agronomic traits that were measured or scored are described in Table 2.

**Table 2.** Agronomic traits evaluated on BC1F5 progenies in 2013 with the units and method of measurement.

| Traits | Abbreviation | Units | Method |
|--------|--------------|-------|--------|
| Seedling vigor | GV | Score (1–9) | Visual score of seedling growth 35 d after sowing from lowest (1) to highest (9) |
| Date to flag leaf appearance | DTFL | Julian day | Number of days from sowing to flag leaf appearance |
| Plant height | PH | cm | Length of the main stalk recorded on 3 random plants from the base of the stalk to the tip of the panicle (maturity). |
| Panicle length | PANL | cm | Length from the first whorl of panicle branches to the tip of the rachis (maturity) |
| Grain yield/m$^2$ | GYLD | g m$^{-2}$ | Threshed grain weight per unit area(maturity) |
| Weight of 100 grains | HGW | g | Weight of 100 grains with grain humidity less than or equal to 12% |

*2.3. Statistical Analysis*

2.3.1. Individual Trial Analysis

Each single trial was analyzed as a separate environment using Model (1) assuming block ~N (0, $\sigma^2$b) and error ~N (0, $\sigma^2$).

Model (1)

$$Y_{ijl} = \mu + G_i + R_j + B_{l(j)} + E_{ijl},$$

where $Y_{ijl}$ is the observed *l*th plot value of the *i*th genotype in the *k*th block within the *j*th replication, $\mu$ is the population mean, $G_i$ is the *i*th genotype, $R_j$ is the *j*th replication, $B_{l(j)}$ is the block within replication, and $E_{ijl}$ is the residual error. Genotypes were considered as random in Model (1) to estimate best linear unbiased predictions (BLUP) for genotype performance and variance components to estimate repeatability ($w^2$) with Model (2) using an adjusted formula for unbalanced data sets [18]. Genotypes were treated as fixed effects to obtain Best Linear Unbiased Estimate the (BLUE) to estimates genotypic means and correlations. Correlations and analyses of variance were conducted using GenStat (14), and BLUEs were computed with BMS (3.0.9). "R" was used for producing box plots.

Repeatability ($w^2$) was estimated as

Model (2)

$$w^2 = \frac{\sigma_g^2}{\sigma_g^2 + \frac{V}{2}},$$

where $\sigma_g^2$ is the genotypic variance, and $V$ is the mean variance of difference between treatment means.

### 2.3.2. Combined Analysis

The linear model used for combined analysis of environments included LP and HP conditions at Samanko location.

Model (3)

$$Y_{ijkl} = \mu + G_i + L_j + GL_{ij} + R(L)_{jk} + B(R(L))_{jkl} + E_{ijkl},$$

where $Y_{ijkl}$ is the observed value of the $i$th genotype in the $l$th block of the $k$th replication of the $j$th environment, $\mu$ the population mean, $G_i$ is the $i$th genotype, $L_j$ is the $j$th environment, $GL_{ij}$ is the interaction of the $i$th genotype and $j$th environment, $R(L)_{jk}$ is the $k$th replication within the $j$th environment, $B(R(L))_{jkl}$ is the $l$th block within the $k$th replication of the $j$th environment, and $E_{ijkl}$ the residual error.

Genotypes and environments were considered as random for estimating variance components used to estimate broad-sense heritability (Model (4)).

Broad-sense heritability ($h^2$) was estimated as

Model (4)

$$h^2 = \frac{\sigma_g^2}{(\sigma_g^2 + \frac{\sigma_{gl}^2}{l}) + (\frac{\sigma_e^2}{rl})},$$

where $\sigma_g^2$ and $\sigma_{gl}^2$ are the components of variance for genotype and genotype by environment interaction over $l$ environments, respectively, and $\sigma_e^2$ is the error variance component over $r$ replications and $l$ environments.

Genetic correlation was estimated using

Model (5)

$$r_G = \frac{r_{\text{phenotpic}}}{\sqrt{(h^2{}_{LP} * h^2{}_{HP})}},$$

where $r_G$ is genetic correlation coefficient of grain yield between HP and LP, $r$ is phenotype correlation, and $h^2$ is repeatability under LP and HP as described.

The effectiveness of indirect (selecting on grain yield under HP conditions) relative to direct selection (selecting on grain yield under LP conditions) for improving LP grain yield ($R_{id}/R_d$) was estimated as

Model (6)

$$R_{id}/R_d = r_G \frac{\sqrt{(h^2{}_{HP})}}{\sqrt{(h^2{}_{LP})}},$$

where $r_G$ is a genetic correlation coefficient of grain yield between HP and LP, and $h^2HP$ and $h^2LP$ are the estimates of repeatability for grain yield under HP and LP conditions, respectively [19].

## 3. Results

### 3.1. Performance for Grain Yield and Related Traits Uunder LP and HP Field Conditions

The overall mean grain yield under LP conditions (116 g m$^{-2}$) was just slightly over half of the level obtained under HP (277 g m$^{-2}$) in 2013 (Table 3). The minimum genotype yields were far inferior to the overall mean under both LP and HP conditions, reflecting the late flowering and inability to fill grain of some progenies. Although there was overlap between low- and HP conditions for progenies with lower yields, numerous progenies produced grain yield under HP that exceeded the highest yield exhibited under LP in 2013. The mean grain yields in 2014 and 2015 under LP conditions (121 and 116 g m$^{-2}$ respectively) were reduced by 57% and 44% relative to the corresponding mean yield under HP (284 and 206 g m$^{-2}$ respectively). The mean dates of flowering under LP conditions were by delayed 2, 5, and 3 d relative to HP in 2013, 2014, and 2015, respectively. Mean plant heights were 295, 331, and 276 cm under HP, whereas under LP, the trials means for plant height were reduced by 36, 102, and 47 cm in 2013, 2014, and 2015, respectively.

### 3.2. Genetic Parameters

#### 3.2.1. Repeatability, Heritability, and Genetic Variance Estimates

The single environment repeatability estimates for grain yield and yield-related traits were high to very high, except for seedling vigor under LP and HP conditions in 2013 (Table 3). The repeatability estimates for grain yield were only slightly higher in the HP relative to LP. This trend was also found for other agronomic traits except for date to flag leaf appearance and seedling vigor.

The variation among progenies over all populations for grain yield and agronomic traits was highly significant ($p < 0.001$) within each P level in 2013 (Table 3). Although the combined analysis of grain yield across P levels for grain yield revealed highly significant ($p < 0.001$) variance components for both genotypes and genotype by P-level interactions, the variance component for genotype was considerably larger than that of genotype by P-level interaction (Table 4). The broad-sense heritability estimate for grain yield was of intermediate magnitude, with flowering (DTLF), plant height (PH), and seed weight (HGW) being higher and seedling vigor (GV) lower (Table 4).

**Table 3.** Components of variance for genotype ($\sigma^2$G) and their standard errors (s.e.); genotype minimum, maximum, and overall mean for genotype best linear unbiased estimates (BLUEs) from Model 3; and repeatability estimates for agronomic traits for 1083 progenies evaluated under low-P (LP) and high-P (HP) conditions in 2013.

| Traits | HP | | LP | | Minimum | | Maximum | | Mean | | Repeatability | |
|---|---|---|---|---|---|---|---|---|---|---|---|---|
| | $\sigma^2$G | s.e. | $\sigma^2$G | s.e. | LP | HP | LP | HP | LP | HP | LP | HP |
| GYLD | 4918 *** | 313 | 1801 *** | 135 | 20 | 40 | 509 | 703 | 166 | 277 | 0.60 | 0.69 |
| DTFL | 56.5 *** | 2.6 | 50.9 *** | 2.3 | 49.5 | 46.5 | 129 | 133 | 82.0 | 80.1 | 0.95 | 0.94 |
| PH | 1801.4 *** | 87.7 | 1185.4 *** | 65.6 | 103 | 115 | 420 | 485 | 256 | 292 | 0.79 | 0.88 |
| HGW | 0.081 *** | 0.004 | 0.091 *** | 0.004 | 0.49 | 0.60 | 3.19 | 3.20 | 2.15 | 2.31 | 0.90 | 0.91 |
| PANL | 11.44 *** | 0.63 | 9.76 *** | 0.57 | 16.0 | 13.5 | 45.5 | 50.0 | 30.0 | 29.3 | 0.76 | 0.79 |
| GV | 0.17 *** | 0.03 | 0.07 *** | 0.02 | 2 | 2 | 8 | 9 | 5.17 | 5.92 | 0.03 | 0.00 |

Growth vigor score (GV), date to flag leaf appearance (DTFL), plant height in cm (PH), panicle length in cm (PANL), grain yield in g m$^2$ (GYLD), and hundred grain weight in g (HGW); G = genotype, s.e. = standard error; * = significance at ($p < 0.05$), ** ($p < 0.01$), *** ($p < 0.001$).

**Table 4.** Variance component estimates of genotype ($\sigma^2$G) and genotype by P-level interaction ($\sigma^2$G × P) with their respective standard error (s.e.) and estimates of broad-sense heritability ($h^2$) from a combined analysis over P levels (Model 3) of 1083 BC1F4 progenies for grain yield and agronomic traits at Samanko in 2013.

| Traits | Combined | | | | |
|---|---|---|---|---|---|
| | $\sigma^2$**G** | **s.e.** | $\sigma^2$**G × P** | **s.e.** | $h^2$ |
| GYLD | 2460 *** | 171.00 | 932 *** | 118.00 | 0.67 |
| DTFL | 52.63 *** | 2.34 | 1.18 *** | 0.19 | 0.96 |
| PH | 1463.2 *** | 69.60 | 39 *** | 15.60 | 0.91 |
| HGW | 0.08 *** | 0.00 | 0.01 *** | 0.00 | 0.91 |
| PANL | 10.12 *** | 0.51 | 0.48 *** | 0.18 | 0.86 |
| GV | 0.10 *** | 0.01 | 0.02 ns | 0.02 | 0.35 |

$\sigma^2$**G** = genotypic variance, s.e. = standard error; * = significance at ($p < 0.05$), ** ($p < 0.01$), *** ($p < 0.001$).

The genotypic variation for grain yield exhibited by the subset of entries over three years (2013–2015) was highly significant ($p < 0.001$) under LP and HP conditions as well as across both P levels (Table 5). The broad-sense heritability estimates were nearly identical under both LP and HP conditions and were only slightly lower than in the single-year (2013) analysis (Table 3). Although the genotype × P-level interaction (G × P) variance component combined over years was highly significant ($p < 0.001$), it was smaller than that of the genotype × year interaction (Table 5). The broad-sense heritability estimate across P levels over years was actually higher than those of the individual P levels (Table 5) and the across P-level estimate with the full set of progenies in 2013 (Table 4).

**Table 5.** Variance component estimates ($\sigma^2$) and corresponding standard error (s.e.) and broad-sense heritability estimates ($h^2$) for 300 progenies under LP, HP, and combined over both P levels across three years.

| Env | Sources of Variation | $\sigma^2$ | s.e. | $h^2$ |
|---|---|---|---|---|
| HP | Genotype | 6319 *** | 1126 | 0.59 |
| | Genotype × Year | 11,570 *** | 1218 | |
| LP | Genotype | 1866 *** | 431 | 0.57 |
| | Genotype × Year | 3726 *** | 478 | |
| Combined | Genotype | 3997 *** | 661 | 0.79 |
| | Genotype × P | 1202 *** | 370 | |
| | Genotype × Year | 4515 *** | 608 | |
| | Genotype × P × Year | 2223 *** | 481 | |

Env = environment, $\sigma^2$ = variance component, s.e. = standard error; * = significance at ($p < 0.05$), ** ($p < 0.01$), *** ($p < 0.001$).

Although the genetic correlations between LP and HP conditions were highly significant for all agronomic traits, the correlation for grain yield was only of intermediate magnitude, whereas values above 0.80 were estimated for traits such as flowering (DTFL), plant height (PH), and seed weight (HGW) (Table 6). Correlations between grain yield and other agronomic traits were significant but weak. These correlations indicated that higher grain yields were associated with earlier flowering (DTFL), larger seed weight (HGW), and taller plant height (PH) in both LP and HP conditions (Table 6). However, under LP conditions, the correlation of yield with flowering (DTFL) was weaker and with seed weight (HGW) was slightly stronger than under HP conditions.

**Table 6.** Genetic correlations (r) of grain yield and agronomic traits of 1083 progenies evaluated under LP and HP in 2013.

| Traits | Correlation (r) | | | | | | | | | | | |
|---|---|---|---|---|---|---|---|---|---|---|---|---|
| | DTFL_HP | DTFL_LP | GV_HP | GV_LP | GYLD_HP | GYLD_LP | PANL_HP | PANL_LP | PH_HP | PH_LP | HGW_HP | HGW_LP |
| DTFL_HP | | | | | | | | | | | | |
| DTFL_LP | 0.91 *** | | | | | | | | | | | |
| GV_HP | −0.16 *** | −0.09 ** | | | | | | | | | | |
| GV_LP | −0.11 *** | −0.25 *** | 0.20 *** | | | | | | | | | |
| GYLD_HP | −0.23 *** | −0.25 *** | 0.44 *** | 0.24 *** | | | | | | | | |
| GYLD_LP | −0.03 ns | −0.13 *** | 0.12 *** | 0.37 *** | 0.54 *** | | | | | | | |
| PANL_HP | 0.05 ns | 0.03 ns | 0.09 ** | 0.06 ns | 0.23 *** | 0.06 ns | | | | | | |
| PANL_LP | 0.19 *** | 0.14 *** | 0.04 ns | 0.05 ns | 0.12 *** | 0.10 ** | 0.73 *** | | | | | |
| PH_HP | 0.29 *** | 0.28 *** | 0.34 *** | 0.12 *** | 0.31 *** | 0.15 *** | 0.44 *** | 0.37 *** | | | | |
| PH_LP | 0.27 *** | 0.19 *** | 0.19 *** | 0.32 *** | 0.22 *** | 0.27 *** | 0.36 *** | 0.43 *** | 0.82 *** | | | |
| HGW_HP | −0.13 *** | −0.14 *** | 0.21 *** | 0.12 | 0.30 *** | 0.25 *** | 0.06 ns | −0.03 ns | 0.13 *** | 0.11 *** | | |
| HGW_LP | −0.05 ns | −0.07 * | 0.14 *** | 0.15 | 0.20 *** | 0.37 *** | 0.07 * | 0.01 ns | 0.18 *** | 0.21 *** | 0.83 *** | |

Growth vigor score (GV), Date to flag leaf appearance (DTFL), Plant height in cm (PH), Panicle length in cm (PANL), Grain yield in g m$^2$(GYLD), Hundred grain weight in g (HGW),* = significance at ($p < 0.05$), ** = ($p < 0.01$), *** ($p < 0.001$), ns = no significant.

3.2.2. Predicted Responses to Direct and Indirect Selection for Grain Yield Under P-Limited Conditions

The genetic correlation for grain yield between LP and HP conditions was 0.81, which, although somewhat elevated, was considerably less than 1.00. The estimates of $R_{id}/R_d$ ratios (Model 6) for the predicted efficiency of indirect (high P) versus direct (low P) selection for grain yield under P-limited conditions were lower than 1.00 in all cases, being 0.87, 0.67, and 0.79 for the 1083 BC1F5 progenies in 2013 and the 298 BC1F5 progenies in 2014 and 2015, respectively.

The progeny mean yields showed very large rank changes between the LP and HP conditions over all 1083 progenies in 2013 and over the 298 selected progenies tested in 2014 and 2015 (Supplementary Materials) Examining the top yielding 10% of progenies in 2013 revealed that only 44 of the 109 (40.4%) top yielding progenies within P-level were common over both LP and HP conditions in 2013. Similarly, only 8 of the 29 (27.6%) top yielding progenies were common under both LP and HP conditions with combined results over 2014 and 2015 (Supplementary Materials)

*3.3. Performance of Specific Populations*

Large variations among progenies developed from a common donor for grain yield, flag leaf appearance, and plant height in 2013 (Figure 3). The selected subset of 298 progenies also showed large variations for grain yield within each population under both LP and HP conditions in 2014 and 2015 (Figure 4).

The population means for grain yield differed over a nearly two-fold range in the LP as well as the HP environments of 2013 (Table 7). Whereas 8 of the 13 populations had mean yields superior to the recurrent parent Lata3 under LP, only two showed numerically superior mean yields under HP. The four populations with the highest mean yields in both LP and HP in 2013 had donor parents of Guinea- (Samba, Soumb) and Caudatum-race (Ribda) from over 1000 km east of Mali (Nigeria and Cameroon) and a Guinea-Caudatum inter-racial line bred in Mali (Grinka). The populations with Guinea-race donors from Mali (N'golo and Douad) or neighboring Burkina Faso (Gnoss) had only intermediate yield levels. The population with the lowest yield performance (Hafid) had a Guinea-magaretiferum variety as donor.

Examining the yields of the subset of progenies evaluated over two years revealed five populations with means superior to the recurrent parent (Lata3) under LP but none under HP (Table 7). The populations with superior mean yields under LP included two populations that also exhibited superior yields in 2013 (Grinka and Soumb) and three other populations (N'golo and Douad with Malian Guinea-race donors and SC566 with a Caudatum-race donor) (Table 7).

The progenies among the top 25% for yield in each population were generally all superior to the recurrent parent under both LP and HP conditions in 2013 (Figure 3) as well as in 2014 and 2015 (Figure 4). Only progenies in the top quartile of the population Hafid did not exceed the recurrent parent (Figure 3), with the top-quartile mean being numerically inferior under both LP and HP conditions (Table 7). Under LP conditions, two populations (Grinka and Soumb) had the highest mean for the top quartile progenies in 2013 as well as combined over 2014 and 2015, with a third population (SC566) ranked third and fifth, respectively (Table 7). The five top-ranking populations for top-quartile progenies means under HP were identical to those under LP in 2013, but in the multiyear evaluation, they included only two of the five populations (Soumb and N'golo) (Table 7).

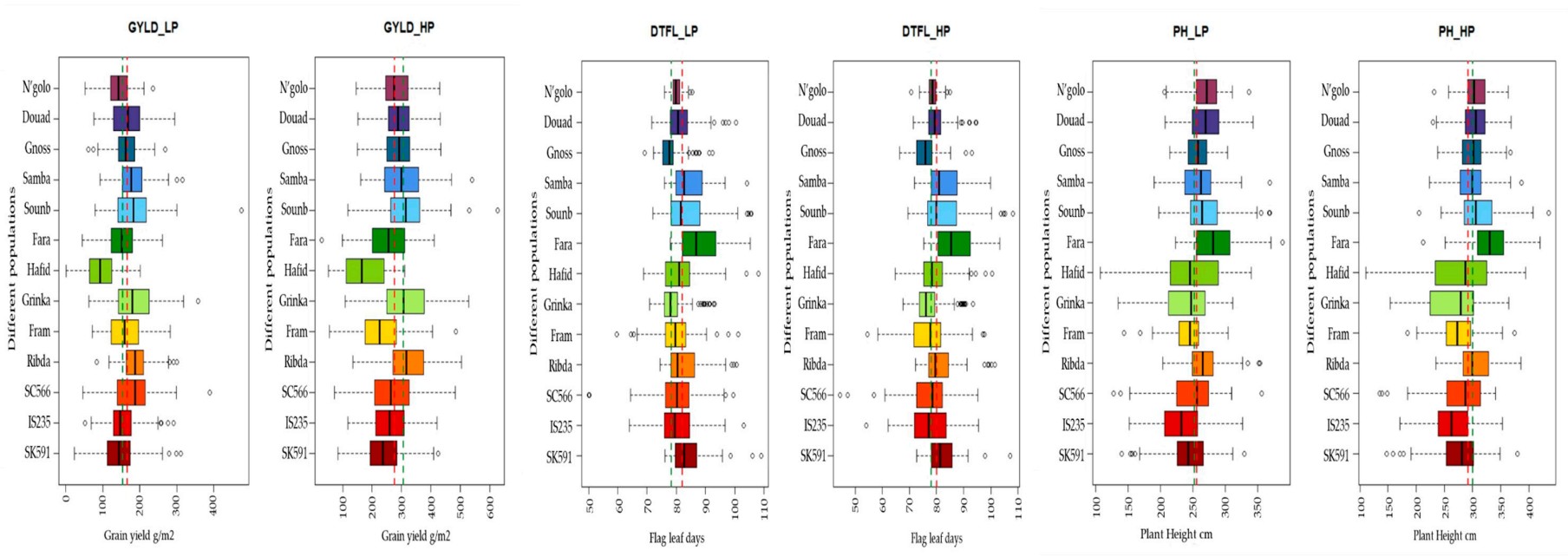

**Figure 3.** Best linear unbiased estimates for grain yield, days to flag leaf, and plant height of progenies, presented as box plots with progenies of a common donor parent, where colored boxes correspond to 25% above and 25% below the median for that group, midlines of each boxplot represent the median, whiskers indicate the total range, circles denote outlier values, and the red- and green-dashed lines indicate the trail and the recurrent parent Lata3 mean yields, respectively, under LP and HP conditions in 2013.

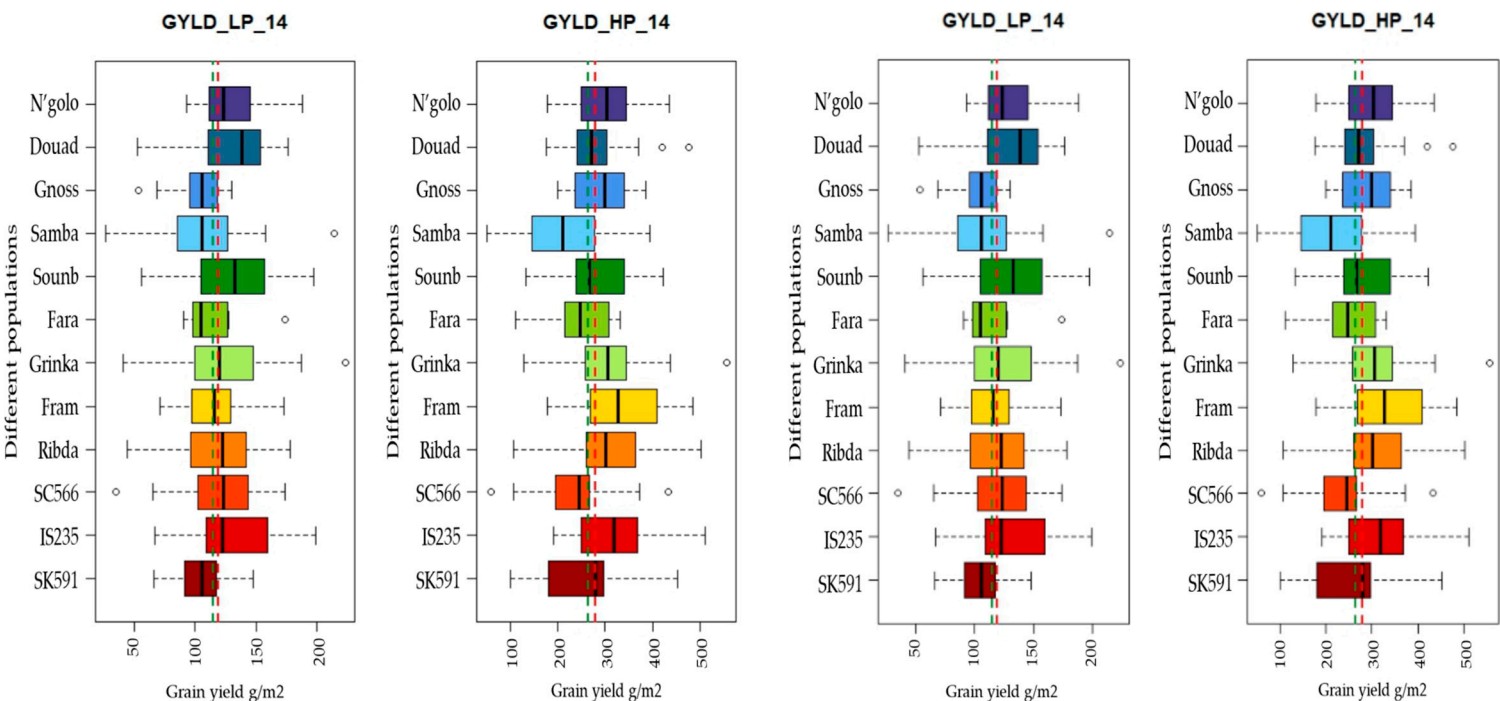

**Figure 4.** Best linear unbiased estimates for grain yield of selected progenies 2013, presented as box plots with progenies of a common donor parent, where the colored boxes correspond to 25% above and 25% below the median for that group, midlines of each boxplot represent the median, whiskers indicate the total range, circles denote outlier values, and the red- and green-dashed lines indicate the trial and the recurrent parent Lata3 mean yields, respectively, under LP and HP conditions in 2014 and 2015.

**Table 7.** Mean grain yields (g m$^{-2}$) of the recurrent parent Lata-3 and of the 13 individual populations based on the Best Linear Unbiased Predictions of individual progeny yields over all- and over the top yielding quartile of progenies with a common donor parent under Low- and High-P conditions over 1083 progenies in 2013 and the 298 progenies in 2014 and 2015.

| Population | 2013 | | | | | 2014 | | | | 2015 | | | |
| | LP | | HP | | # Progenies Selected | LP | | HP | | LP | | HP | |
| | Overall | Top Quartile | Overall | Top Quartile | | Overall | Top Quartile | Overall | Top Quartile | Overall | Top Quartile | Overall | Top Quartile |
|---|---|---|---|---|---|---|---|---|---|---|---|---|---|
| N'golo | 143 | 186 | 284 | 358 | 26 | 128 | 159 | 300 | 385 | 109 | 148 | 209 | 258 |
| Douad | 165 | 227 | 292 | 369 | 29 | 131 | 158 | 280 | 350 | 102 | 149 | 205 | 221 |
| Gnoss | 164 | 212 | 291 | 369 | 16 | 103 | 126 | 292 | 361 | 96 | 126 | 191 | 245 |
| Samba | 184 | 240 | 304 | 399 | 39 | 105 | 146 | 220 | 324 | 85 | 133 | 191 | 255 |
| Soumb | 186 | 260 | 315 | 421 | 36 | 129 | 170 | 286 | 378 | 109 | 152 | 225 | 276 |
| Fara | 152 | 202 | 257 | 352 | 11 | 114 | 143 | 251 | 324 | 85 | 115 | 194 | 240 |
| Hafid | 100 | 150 | 176 | 264 | 0 | - | - | - | - | - | - | - | |
| Grinka | 186 | 268 | 308 | 426 | 47 | 123 | 171 | 304 | 401 | 104 | 159 | 214 | 289 |
| Fram | 162 | 230 | 236 | 343 | 15 | 116 | 149 | 332 | 458 | 88 | 123 | 170 | 218 |
| Ribda | 190 | 241 | 318 | 419 | 32 | 121 | 159 | 311 | 416 | 92 | 132 | 209 | 277 |
| SC566 | 179 | 249 | 263 | 375 | 22 | 122 | 159 | 243 | 328 | 106 | 141 | 181 | 217 |
| IS235 | 155 | 215 | 261 | 350 | 12 | 132 | 191 | 319 | 427 | 79 | 117 | 199 | 259 |
| SK591 | 151 | 226 | 241 | 332 | 13 | 105 | 131 | 252 | 347 | 101 | 141 | 176 | 213 |
| Lata3 | 155 | | 308 | | | 114 | | 375 | | 106 | | 202 | |

(#) it mean number (Number of progenies selected in 2013 under LP and HP at ICRISAT Mali, samanko station) for their evaluation under LP and HP in 2014 and 2015.

## 4. Discussion

The considerable and significant reduction of grain yield and plant height under LP, and the delay in heading under LP conditions relative to HP (Table 3), suggest that the field conditions in this study were appropriate for investigating selection strategies for genetic improvement of grain yield under contrasting P conditions. Such yield reductions due to LP have been previously reported by numerous other studies [6,8,20–24]. Furthermore, the acceptable repeatability estimates for grain yield in both LP and HP conditions (Table 3) give confidence in the results obtained in this study.

### 4.1. Genetic Parameters

The significant genetic variation and the acceptable and nearly identical broad-sense heritabilities for grain yield under both LP and HP conditions, estimated over multiple years (Table 5), suggest that selection for grain yield among these backcross progenies should be effective under either LP or HP levels. Furthermore, varietal development efforts targeting P-deficient production environments is expected to make greater genetic gains through direct selection for yield under LP conditions, as indicated by the $R_{id}/R_d$ ratios lower than 1.00. Leiser et al., 2012, came to the same conclusion, reporting quite similar $R_{id}/R_d$ ratios from a panel of West African sorghum varieties evaluated under LP and HP conditions over multiple years. Studies on selection for grain yield under contrasting nitrogen (N) levels also reported direct selection under low N to be more effective when targeting production systems with low soil N [19,25].

### 4.2. Usefulness of BCNAM Populations

The recurrent parent (Lata3) used to create the backcross progenies evaluated in this study is the male parent of high-yielding Guinea-race sorghum hybrids [26,27], including "Pablo", one of the most widely cultivated hybrids in Mali [26]. The BCNAM populations in this study thus represent promising material for diversifying the male parent pool, as their genetic backgrounds are expected to be approximately 75% from Lata3 and 25% derived from the donor parents, which are very diverse and most identified to be restorer lines. The yield superiorities under LP conditions of several of these BCNAM populations and individual backcross progenies relative to Lata3 (Table 7 and Figures 3 and 4) thus indicates considerable potential for making genetic gains for, per se, yield performance of new male parents targeting the predominant low-input production systems of Mali and West Africa. The report that male parent yield performance under LP conditions was positively related to sorghum hybrid yield in P-deficient environments in Mali (correlations of 0.41 to 0.85, average of 0.59) [15] highlights the potential contribution these high-yielding BCNAM progenies could make to hybrid development for P-limited environments.

The two BCNAM populations with the highest yielding backcross progenies under LP conditions across multiple years (Grinka and Soumb) (Table 7) had donor parents (Grinkan and IS 15401, respectively) that were previously identified to be among the top yielding entries across a panel of 70 West African varieties, with IS 15401 exhibiting specific adaptation to LP environments and Grinkan among the top-ranked varieties for yield under both LP and HP conditions (Leiser et al. 2012). The superior yields of progenies from both the Soumb and Grinka populations under HP as well as LP conditions (Table 7) suggests that genes for productivity other than or in addition to those for adaptation to LP may have been contributed by these donors.

A combining ability study revealed that male parents with introgression of IS 15401 or Ribdahu exhibited a superior general combining ability (GCA) when crossed onto newly developed Malian seed parents (Kante et al. 2019). This study observed a trend of higher GCA associated with male parents having introgressed germplasm from the more humid sorghum growing regions of Cameroon and Nigeria. Our study also showed that introgression of some sorghum accessions from that region can create useful variation for grain yield under LP conditions, as exhibited by the Soumb, Ribda,

and Samba populations, whereas other donors did not, with the SK591 and Fara populations having inferior yields (Table 7).

Several of the donor parents used to create the BCNAM populations (IS 15401, Ribdahu, Sambalma) are actually late maturing and unadapted to the major sorghum belt of Mali, originating in more humid regions over 1000 km east of Mali. Despite the poor adaptation of many our BCNAM donors to the Malian environments, the yield superiority of many BCIF5 progenies relative to the elite recurrent parent and the large variation for yield indicates that useful genetic variation can be obtained through the BCNAM approach used here. This approach, based on use of an elite recurrent parent, crossing to a range of diverse donors, and conducting a single backcross to the elite parent and advancement of many BC1F1 derivatives with only limited early generation progeny culling for critical adaptation traits (such as maturity), was pioneered for diversifying sorghum breeding material in Australia (Jordan et al. 2011).

## 5. Implications and Conclusions

Farmers in West Africa predominantly cultivate sorghum under low-fertility and, particularly, LP conditions [3,5,6,9]. For sorghum breeders to maximize genetic gains for grain yield under LP conditions in West Africa, direct testing and selection under LP conditions was shown to be feasible and necessary by this study and others [6]. Sorghum breeding programs in West Africa will, therefore, need to manage certain research station fields for LP fertility that better represent farmers' soil conditions or work with farmers to conduct certain activities directly in farmers' LP fields. Both approaches are feasible, as was shown by results of this study and that of [3]. The diversification of Malian breeding materials using the BCNAM approach for introgressing diverse germplasm, including sorghums from the more humid regions of Nigeria and Cameroon, can create useful genetic variation for improving grain yield under LP conditions. The materials generated in this study appear to be highly promising for diversifying the male parent pool for sorghum hybrids in Mali. Nevertheless, the genetic parameters estimated here show that use of conventional selection methods should be feasible for these traits under both LP and HP conditions.

**Supplementary Materials:** The following are available online at http://www.mdpi.com/2073-4395/9/11/742/s1.

**Author Contributions:** Conceptualization, C.D., E.W. and H.F.W.R.; methodology, C.D. and W.L.; validation, I.S. and C.D. and B.N.; formal analysis, C.D. and W.L.; investigation, C.D., M.S.; and B.S.; data curation, C.D. and I.S.; writing—original draft preparation, C.D.; writing—review and editing, H.F.W.R. and V.G. and A.T.; supervision, A.T. and E.Y.D. and D.K.D.; project administration, B.N. and E.W.; funding acquisition, E.W

**Funding:** This research received no external funding.

**Acknowledgments:** The authors thank the technical staff of the sorghum-breeding program in ICRISAT and IER-Mali for their contribution in phenotyping. We thanks the team of West Africa Center for Crop Improvement (WACCI) and AGRA project funded by the foundation bill and Melina gate for their participation. This work was undertaken as a part of the CGIAR Research Program on Dryland Cereals.

**Conflicts of Interest:** The authors declare there to be no conflict of interest.

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
