# Peer review of "Genetic Diversification and Selection Strategies for Improving Sorghum Grain Yield Under Phosphorous-Deficient Conditions in West Africa"

_agronomy, doi:10.3390/agronomy9110742_

Round 1
Reviewer 1 Report
Dear authors,
my suggestions to the editors are as follows:
the manuscript deals with widening the genetic base of Malian sorghum, with special focus on low P environments. The authors show the usefulness of a backcrossing approach established for sorghum breeding in Australia also for Sahelian environments. Hence, it is a nice proof-of-concept study with high value for local breeding. The quality of presentation and soundness are fine. Table 6 could be modified and presented as a heatmap.
The paper provides relevant information and recommendations for sorghum breeding in low-input agriculture prone to phosphorus deficiency in Sahelian Africa. The breeding procedure which the authors apply was established and published by sorghum breeders in Australia, so it is rather a proof-of-concept study than a novel discovery. Nonetheless, I consider it of high value for the plant breeding community, especially in regard to the particular environment on which it is focused. It is well written and the conclusions are supported by the presented results. From my point of view, there is almost nothing to critize. The correlation matrix could be displayed as heatmap, but this is only a minor remark.
Author Response
Thank you for your comments and suggestion, we appreciated, I tried to modified the table table 6 to correlation heatmap, but I got some problem with "R" to install ggplot2 package in my machine, that why I was not able to modify the table.

Reviewer 2 Report
Here are my comments to the authors:
figure 1: please revise the legend is Bc1F4 or Bc1F5?
plant materials section: please add some details on field conditions applied during the evaluation and the development of populations. Have you tried different environments?
line 141. please correct identicial in identical
table 1: double dot in the legend
table 3: I cannot see the significance in the table
table 4: please insert the mining of significance in a foot note
table 5: change the graphic of this table
table7: please report data for each single year separately
paragraph 3.1 is too short, revise the distibution of paragraph in result section.
abstract and text: in the abstract you clearly report some data referred to GBS technique (lines 29-30), but in the text there is no comment, details or others about this. So, please remove this part in the abstract, or otherwise please comment and report results about this, both in material and methods that in results.
Author Response
Thank you for your comments and suggestion, We appreciated.
figure 1: please revise the legend is Bc1F4 or Bc1F5? (We corrected BC1F4)
line 141. please correct identicial in identical ((We corrected)
table 1: double dot in the legend (We corrected BC1F4)
table 3: I cannot see the significance in the table ((We corrected)
table 4: please insert the mining of significance in a foot note ((We corrected)
table 5: change the graphic of this table (We modified the table)
table7: please report data for each single year separately ((We corrected )
paragraph 3.1 is too short, revise the distibution of paragraph in result section. (we mdified the this paragraph)
abstract and text: (We removed the part tht referred to GBS technique (lines 29-30))
some detail have been added on field conditions applied during the evaluation and the development of populations.
We have used one location under LP and HP; but the initial plan was two, but get some diffuculties implanted at the second location.
Thank you.
